# Flower-like Thiourea–Formaldehyde Resin Microspheres for the Adsorption of Silver Ions

**DOI:** 10.3390/polym15112423

**Published:** 2023-05-23

**Authors:** Yuhan Li, Xiaoli Wang, Jing Xia, Guangwei Zhou, Xiaomu Wang, Dingxuan Wang, Junying Zhang, Jue Cheng, Feng Gao

**Affiliations:** 1Key Laboratory of Carbon Fiber and Functional Polymers, Ministry of Education, Beijing University of Chemical Technology, Beijing 100029, China; chn_liyh@163.com (Y.L.); zgwbuct@163.com (G.Z.); w343002449@163.com (X.W.); wangdx_2022@163.com (D.W.); zhangjy@mail.buct.edu.cn (J.Z.); 2Aerospace Research Institute of Materials & Processing Technology, Beijing 100076, China; wangxiaoli@arimt.com; 3Ansteel Beijing Research Institute Co., Ltd., Beijing 102200, China; xiaj0001@126.com

**Keywords:** silver ion adsorption, thiourea–formaldehyde resin, morphology control

## Abstract

Around a quarter of annual worldwide silver consumption comes from recycling. It remains a primary target for researchers to increase the silver ion adsorption capacity of the chelate resin. Herein, a series of flower-like thiourea–formaldehyde microspheres (FTFM) possessing diameters of 15–20 μm were prepared via a one-step reaction under acidic conditions, and the effects of the monomer molar ratio and reaction time on the micro-flower morphology, specific surface area, and silver ion adsorption performance were explored. The nanoflower-like microstructure showed the maximum specific surface area 18.98 ± 0.949 m^2^/g, which was 55.8 times higher than that of the solid microsphere control. As a result, the maximum silver ion adsorption capacity was 7.95 ± 0.396 mmol/g, which was 10.9 times higher than that of the control. Kinetic studies showed that the equilibrium adsorption amount of FT_1_F_4_M was 12.61 ± 0.016 mmol/g, which was 11.6 times higher than that of the control. Additionally, the isotherm study of the adsorption process was performed, and the maximum adsorption capacity of FT_1_F_4_M was 18.17 ± 1.28 mmol/g, which was 13.8 times that of the control according to the Langmuir adsorption model. Its high absorption efficiency, convenient preparation strategy, and low cost recommend FTFM bright for further use in industrial applications.

## 1. Introduction

Growing industrialization and urbanization resulted in severe pollution, particularly that of the aquatic environment [1]. The wastewater produced by the industrial society contains abundant toxic heavy metals such as gold, silver, and cadmium [2,3]. Silver is a noble metal that is widely used in the photography, electronics, chemical, and jewelry industries [1,4,5]. Studies showed that around a quarter of the silver produced from the industry was from wastewater recycling [2,6]. In recent decades, various silver ion recycling strategies have been proposed and evaluated, such as adsorption [1], sedimentation [7], precipitation [8], ion exchange [9], nano-filtration [10], and electrolysis [11]. Among these methods, the adsorbents used for silver recycling have received increasing attention due to their low cost [5]. Common adsorbents include chelating resins [12], activated carbon [5,13], and other biomimetic materials [14].

Polymeric materials containing sulfinyl groups can be used as silver adsorbents due to the high binding affinity between the carbon–sulfur double bond (-C=S-) and silver ion according to hard-soft-acid-base theory [15,16]. Thiourea–formaldehyde resin is a typical chelating resin containing abundant sulfinyl groups that is commonly used for silver ion adsorption [1,5,17,18]. Kirci et al. indicated that thiourea–formaldehyde resin had higher selectivity to silver ion compared with Cu^2+^ or Zn^2+^ and a maximum adsorption capacity for silver ions of 0.539 mmol/g [19]. Yirikoglu et al. separated silver from base and earth metal ions using a melamine–formaldehyde–thiourea (MFT) chelating resin [20]. It was concluded that chelation governed the adsorption mechanism of silver ions, and the maximum adsorption capacity was 0.556 mol/g. Iglesias et al. investigated the silver adsorption capacity of chelamine with three different resins with thiol chelating groups: Duolite GT-73, Purolite Thiomethyl resin, and Spheron Thiol [21]. The effectiveness of the resins in adsorbing silver was ordered below: Purolite Thiomethyl resin (2.79 mmol/g) > Duolite GT-73 (1.48 mmol/g) > Spheron Thiol (1.11 mmol/g). This order indicated that the chelate resin has excellent silver ion adsorption performance.

Besides the backbone polymer structural optimization, the design of the adsorbent micromorphology was also an applicable strategy for increasing the adsorption capacity. Hao et al. prepared thiourea–formaldehyde silver absorbent with a controllable porous structure, and the results indicated not only the increasing specific area, but also the larger porous size was able to boom the silver absorption process [22]. Hou et al. designed a shell-hole structural that increases the silver ion adsorption capacity of silver ion-imprinted particles to 0.745 mol/g [23]. Zhao et al. coated Fe_3_O_4_ nanoparticles on the surface of MOF-525 (metal–organic framework) to form Fe_3_O_4_@MOF-525 via the secondary growth method [24]. Compared with the parent MOF-525 (236 m^2^/g), the Brunauer–Emmett–Teller (BET) surface area of Fe_3_O_4_@MOF-525 is significantly increased (427 m^2^/g), which resulted in enhanced adsorption performance. Moreover, the maximum adsorption capacities of tetracycline (TC) and diclofenac sodium (DF) on Fe_3_O_4_@MOF-525 reached 0.62 and 2.34 mol/g, respectively. Zhai et al. fabricated the three-dimensional flower-like magnetic CoFe_2_O_4_/CoFe-LDHs through the hydrothermal method [25]. The microflower presented a highly porous open-space three-dimensional (3D) structure, and the maximum adsorption capacity of the composite for Orange II was 5.93 mol/g. Wu et al. dissolved polypyrrole and polyethersulfone in N-methyl-2-pyrrolidone (NMP), preparing nano-fibrous membranes by electrospinning for silver ion adsorption [26]. The diameters of the synthesized nanofibers ranged from 410 to 540 nm with enhanced porosity and surface area, providing abundant metal binding sites and resulting in a maximum absorption capacity of 0.33 mol/g for silver ions. Jiang et al. prepared porous zinc oxide from two-dimensional nanosheets by the hydrothermal method, enlarging the specific surface area up to 57 m^2^/g, and the corresponding adsorption capacity for Congo red reached 0.48 mol/g, which was 463.89% higher compared with the commercially available porous microspheres [27].

It was discovered in our previous work that urea formaldehyde nanosheets were constructed by adjusting the reaction pH, the monomer ratio, and the reaction temperature [25]. The formed nanosheets were then assembled generating flower-like nanospheres (UFMs). The thickness of the ‘petals’ in the UFMs ranged from 30 to 50 nm, and the flower-like structure endowed UFMs the large specific surface area [28]. The design presented by this work was inspired by the above results. The thiourea and formaldehyde reacted forming thiourea–formaldehyde resin via the Mannich reaction, which was similar to the reaction between urea and formaldehyde. We synthesized and characterized a series of flower-like thiourea–formaldehyde microspheres (FTFM) possessing diameters of 15–20 μm via a one-step reaction under acidic conditions and studied the effect of specific surface area on adsorption capacity. Then, we explored the relationship between adsorption capacity and specific surface area. It was hypothesized that (1) a flower-like microsphere was able to be generated by the reaction between thiourea and formaldehyde (FTFM), which was able to adsorb silver ions by chelation; (2) the structure of FTFM was tunable by adjusting the monomer molar ratio; (3) the flower-like structure in the micro scale was able to dramatically increase the specific surface areas of the thiourea–formaldehyde resin, which would enhance the silver ion adsorption efficiency.

The FTFM was synthesized, characterized, and used as a silver ion adsorbent for the first time. Variety types of thiourea–formaldehyde resins were synthesized with different reaction conditions to verify the raised hypothesis (Figure 1). The chemical structures of the formulations were characterized by Fourier transform infrared (FTIR) spectroscopy measurements. Additionally, the morphology of the resin microspheres was examined by scanning electron microscopy (SEM), and the specific surface area of each formulation was tested according to BET test. The silver ion adsorbent efficiency was calculated based on ultraviolet (UV)–visible (vis) spectra of the adsorbent suspension after adsorption; also the corresponding adsorption kinetic was studied. Furthermore, the relationship between the synthesis conditions and micromorphology was constructed, and the corresponding affection towards the silver ion adsorbent efficiency was evaluated and discussed. The microflower was synthesized via a one-step reaction in aqueous environment, followed by simple rinsing and filtration after precipitation. The microflowers showed a high specific surface area, which significantly enhanced the silver ion adsorption efficiency. Its simple preparation, low cost, and high efficiency give FTFM a bright future in industrial applications.

## 2. Experimental

### 2.1. Preparation of FTFM

Thiourea (T) (Tianjin Damao Chemical Reagent Factory, Tianjin, China) and formaldehyde (F) (37~40%, Beijing Innochem Technology Co., Ltd., Beijing, China) were reacted together to form a thiourea–formaldehyde resin via the Mannich reaction. Deionized (DI) water (100 mL) was charged into a 150 mL beaker, and thiourea and formaldehyde were added. The molar ratio of thiourea to formaldehyde was listed in Table 1 along with the corresponding sample abbreviations. The pH value of the mixture was adjusted to 2.0 by formic acid (≥98%) and ammonia (28~30%) (Beijing Bailingwei Technology Co., Ltd., Beijing, China) at 25 ℃. The solution was then stirred in a water bath at 50 °C for 4 h, and a white precipitate appeared at the bottom of the beaker, which was FTFM. FTFM was then collected by filtration and washed with DI water 3 times to remove the unreacted reagents.

Solid thiourea–formaldehyde microspheres (STFM) were fabricated by adding NH_4_Cl (Shanghai Macklin Biochemical Co., Ltd., Shanghai, China) to the mixture after adjusting the pH to 2. The white precipitation was generated after stirring for 4 h. The molar amount of the NH_4_Cl and the corresponding abbreviations are listed in Table 1. STFM was collected by filtration (filter paper specification: nylon—66, 0.45 μm) and then washed 3 times with DI water (100 mL).

### 2.2. Characterization of the FTFM

The morphology of FTFM was characterized by SEM. A small amount of the sample was dispersed in aqueous solution, a few drops were placed onto the silicon wafer, and the sample was dried with infrared light. The surface was sprayed with gold (10 mA) by an Oxford Quorum SC7620 sputter coater for 45 s under vacuum. The surface morphologies of the samples were obtained using SEM (TESCAN MIRA LMS, Beijing Yake Chenxu Technology Co., Ltd., Beijing, China) operated at an accelerating voltage of 3 kV. The surface element distribution of the samples was tested by energy dispersive X-ray spectroscopy (EDS), and the accelerating voltage was set to 15 kV. The structure of FTFM was characterized by FTIR measurements. Dry FTFM powder (approximately 1 mg) and dry KBr (approximately 200 mg) were mixed and grinded for 1 min. The mixture was then placed on the tablet press with a pressure of 15 MPa for 30–60 s to obtain a transparent tablet. The FTIR spectra of the samples were recorded using a Fourier transform interferometer (ALPHA-T, Bruker, Germany) in the range 400–4000 cm^–1^. The special surface area was observed by BET analysis. The samples were pretreated under vacuum at 120 °C for 7 h in the standard degassing station. Testing was performed by a fully automated specific surface area analyzer (APSP 2460, Micromeritics, Norcross, GA, USA). The porosity of the sample surface was characterized through nitrogen adsorption–desorption isotherm analysis at 77 K, and the surface areas were calculated by the BET equation. The silver ion adsorption capacity was obtained by UV absorption test. After reaching the specified adsorption time, the microspheres in the solution were removed with a disposable needle filter (pore size 0.45 μm, MCF film). Additionally, the UV absorption maximum corresponding to silver ions was determined by measuring the absorbance of the supernatant at 400 nm using a UV−vis spectrophotometer (Lambda 35, PerkinElmer).

### 2.3. Calculation of Adsorption Capacity

The FTFM (0.1g) with a different monomer ratio was placed with 25 mL 0.1mol/L AgNO_3_ solution for 20 min. After adsorption for 20 min, 1 mL of the suspension was subjected to UV absorption measurements (Hitachi U4150, Beijing, China). The concentration of remaining silver ions in the solution was measured, and the monomer ratio with the maximum adsorption capacity was obtained. Furthermore, the optimal monomer ratio that yielded the maximum adsorption capacity was obtained. These data are presented in the supporting information [29].

### 2.4. Adsorption Kinetics

The FTFM adsorbent (0.1 g) was added to 25 mL of 0.1 mol/L AgNO3 solution (pH = 6.0) in a centrifuge tube, and adsorption was allowed to occur for 10–480 min. Samples of 5 mL solution were withdrawn at scheduled time intervals and analyzed for silver ion concentration. Based on Brandani’s research on adsorption kinetics [30], the obtained data were fitted to pseudo-first-order (PFO) [31] and pseudo-second-order (PSO) [32] models.

### 2.5. Adsorption Isotherms

At the optimal pH of 6.0, a solution of silver ion (25 mL, 20–100 mmol/L) and FTFM (0.1 g) were placed in a centrifuge tube. The obtained data were fitted to the Langmuir model. The Langmuir model assumes that the adsorption process occurs in a single layer on a homogeneous medium [33].

## 3. Results and Discussions

The molar ratio between thiourea and formaldehyde varied in order to optimize the morphology by controlling the self-assembly process. The formulation details and abbreviations for each sample set were illustrated in Table 1. Figure 1 shows SEM images of the particles obtained at different reaction times. For FT_1_F_1_M, flower-like microspheres with a diameter of 10 µm were obtained after 1 h (Figure 1b). However, fractures were presented in the sample, indicating the incompletion of the self-assembly process (Figure 1a). The amount of fractures reduced significantly when the reaction time reached 2 h, as presented in Figure 1c, and the diameter of the FTFM was around 15 μm (Figure 1d), which was significantly higher than that of the sample of 1 h reaction. The fractures of the thiourea–formaldehyde resin were hardly observed when the reaction time increased to 4 h, as shown in Figure 1e. Additionally, the corresponding diameter of FTFM remains approximately 15 μm, showing no significant difference compared with that of the sample after 2 h reaction. The amount and distribution of the resin fractures showed no significant change when the reaction time increased to 8 h (Figure 1g), and the diameter of FTFM kept almost unchanged, which was around 15 μm, as presented by Figure 1h. The results indicated that the reaction between formaldehyde and thiourea and the corresponding self-assembly process was completed when the reaction time exceeded 4 h, and the extent of the reaction time showed no significant influence on the diameter of the FTFM.

The effect of the thiourea-to-formaldehyde ratio on the particle morphology was explored, which was presented in Figure 2. As mentioned previously, the reaction time was set to 4 h. The diameter of FT_1_F_0.5_M was around 15 μm, as demonstrated in Figure 2a. The petal size decreased significantly when the formulations came to FT_1_F_1_M, as presented in Figure 1b, and the diameter of the micro-flower showed no significant change. The diameter of the FTFM kept around 15 μm in FT_1_F_2_M, as shown in Figure 2c. The petal size decreased and the petal density increased significantly, compared with that of FT_1_F_1_M. Up to increasing the molar ratio further (FT_1_F_4_M; see Figure 2d), the petal size continues to decrease, and the petal density increases. At the same time, the diameter showed no significant change. The petals on the FTFM became smaller and were distributed more compactly when the molar ration between thiourea and formaldehyde increased from 1:0.5 to 1:4. The self-assembly mechanism of the FTFM was not totally clear. The proposed reason was that the self-assembly was accelerated when the proportion of the formaldehyde increased due to the increasing number of hydrogen bonds. In addition to the formation of the microflowers, solid microspheres were formed by the addition of ammonia chloride; the probable reason for this was that the Mannich reaction between thiourea and formaldehyde was suppressed, further reducing the size or even preventing the formation of petals. Figure 2c demonstrated the morphology of S_1_T_1_F_4_M, in which the molar ratio between ammonia chloride and thiourea was 0.0187:1. The petal size of S_1_T_1_F_4_M significantly reduced and the petal density increased compared with that of FT_1_F_4_M. At the same time, the solid microspheres appeared. When the ratio between ammonia chloride and thiourea increased to 0.0374:1 (S_2_T_1_F_4_M), no microflowers were observed (Figure 2f). The results indicated that the variation of the monomer ratio may dominate the petal size and density for FTFM. However, the corresponding influence of times towards the FTFM dimension was not significant. The introduction of ammonia chloride led to the formation of thiourea–formaldehyde microspheres with a diameter ranging from 8–10 μm, which could be used as the control to investigate the importance of the microflower morphology for silver ion adsorption.

The specific surface area of the FTFM with different formulations was evaluated in order to verify the proposed hypothesis. Figure 3 demonstrated that the BET specific area of the solid thiourea–formaldehyde microsphere (S_2_T_1_F_4_M) was 0.34 ± 0.017 m^2^/g. This value was increased more than 20 times to 7.46 ± 0.373 m^2^/g when the formulation changed to FT_1_F_0.5_M where the flower-like morphology was formed, which indicated that the self-assembly of the thiourea–formaldehyde petals was able to dramatically increase the specific surface area compared with that of the solid spheres. The specific surface area increased to 14.18 ± 0.709, 16.15 ± 0.808, and 18.98 ± 0.949 m^2^/g for FT_1_F_1_M, FT_1_F_2_M, and FT_1_F_4_M, respectively. The results were consistent with the morphology evaluation, i.e., with an increase in the thiourea-to-formaldehyde molar ratio, the petal size decreased and the petal density increased.

The silver ion adsorption capacity was evaluated via the suspension UV-vis spectrum after adsorption. Silver ion has the maximum UV absorption at 303 nm as presented in Figure 4a, and the UV extinction coefficient of silver ion at 303 nm was 6.9 × 10^3^ cm^2^/mol. The corresponding calculation and fitting information were presented in the Appendix A. For the adsorption experiments, 0.15 g of FT_1_F_4_M was added to 25 mL 0.1 mol/L AgNO_3_ solution. As shown in Figure 4a, the UV absorption peak at 303 nm reduced significantly after 10 min of adsorption, indicating that FTFM was able to reduce the concentration of silver ions in the solution. The amount of residual silver ion kept decreasing with the adsorption time, and the corresponding adsorption capacity was calculated and presented in the Appendix A. Figure 4b concludes the silver ion adsorption capacity of FTFM with different formulations after 20 min incubation in the AgNO_3_ solution. The maximum adsorption capacity of the solid spheres was 0.73 ± 0.037 mmol/g, which increased significantly to 4.02 ± 0.201 mmol/g for FT_1_F_0.5_M and increased further to 5.52 ± 0.276, 6.96 ± 0.348, and 7.95 ± 0.398 mmol/g for FT_1_F_1_M, FT_1_F_2_M, and FT_1_F_4_M, respectively. The adsorption capacity result was consistent with the morphology and BET specific area that the close-packed microflower endowed the microspheres with a larger specific surface area, increasing the interaction between the silver ions and the thiourea–formaldehyde resin. Subsequently, the morphology and element composition of FT_1_F_4_M after 20 min adsorption were examined via SEM-EDS. The microsphere kept its flower-like shape after the adsorption, and the diameter showed no significant change, i.e., it remained at approximately 15 μm. However, the petals became significant thicker due to the stacking of silver on the surface. Figure 4d–g shows the element distribution on the surface of the FTFM microsphere after 20 min adsorption, and the silver atoms were uniformly distributed on the surface of the microflower. The homogeneous distribution of the N and S atoms indicated that the thiourea–formaldehyde structure was not disturbed by the adsorption. Among them, silver atoms account for 68.68 wt% of the surface elements (Appendix A).

The chemical structure of the FTFM before and after the adsorption was further verified by the FTIR, and the corresponding spectrum was plotted in Figure 5. The peaks located at 3311 and 1541 cm^−1^ were assigned to the N-H stretching and bending vibrations, respectively, which were greatly weakened after the thiourea–formaldehyde resin was formed. Additionally, this trend became more prominent when the molar ratio between thiourea and formaldehyde increased from 1:0.5 to 1:4 as more N-H was replaced by N-C. Correspondingly, the peaks located at 1327 cm^−1^ representing the N-C connection kept increasing during this process. The peaks caused by the vibration of N-H became wide and unsignificant for the solid microspheres (S_2_T_1_F_4_M) quenched by ammonia chloride. Figure 5b demonstrates the FTIR spectrum of the FTFM samples after 20 min adsorption at 0.1 mol/L AgNO_3_ solution under room temperature. The peaks located around 1403 cm^−1^ were assigned to C=S from the thiourea and were greatly reduced after the absorption in the AgNO_3_ solution, which was supposed to be caused by the chelation between C=S and silver ions. The peaks located around 3311 and 1541 cm^−1^ were broadened and weakened after adsorption, indicating the chelation between silver ions and N-H during the adsorption. The change of C=S structure of the solid microspheres (S_2_T_1_F_4_M) was hardly observed from the FTIR spectrum; however, the reduction in the peak around 1541 cm^−1^ indicated the chelation between silver ions and N-H on the surface of the solid microspheres. The FTIR spectrum confirmed the Mannich reaction between formaldehyde and thiourea and the chemical structure evolution of different formulations. Additionally, the chelation of NH-Ag^+^ and C=S-Ag^+^ was confirmed for all flower-like microspheres.

The adsorption kinetics of FTFM to silver ions were studied. Figure 6 illustrates the relationship between the silver ion adsorption capacity of the thiourea–formaldehyde microspheres and time. The solid thiourea–formaldehyde spheres were able to adsorb silver in aqueous environment, and the capacity increased with the incubation time, which was much lower than that of the flower-like microspheres due to the low specific surface area. The silver ion adsorption capacity of FTFM increased with the raise of the molar ratio between the formaldehyde and the thiourea. The results of the kinetic study were consistent with those of the specific surface area. It was clearly reflected by Figure 6a that the increasing speed of the adsorption capacity slowed down when the adsorption time exceeded 120 min and showed no significant change after 240 min. This indicated that the adsorption and de-adsorption process on the petal surfaces reached equilibrium when the incubation time exceeded 240 min. The pseudo-first order fitting was applied to the adsorption process of FTFM according to Equation (1) presented in Table 2. The values of ln(Q_e_-Q_t_) and time are linear fitted and plotted in Figure 6b, where Q_e_ is the experimental equilibrium adsorption capacity whose value is 240 min, and Q_t_ refers to the adsorption capacity when time is t. The coefficient of determination (R^2^) of the pseudo-first-order fitting is presented in Table 2, ranging from 0.845 to 0.949. The calculated equilibrium adsorption capacity of the pseudo-first-order fitting (Q_e1_) from the intercept of the linear fitting is illustrated in Table 2, ranging from 0.53 ± 1.223 to 3.55 ± 2.061 mmol/g, significantly lower than the experimental values.

Moreover, the pseudo-second-order fitting of the adsorption process is plotted in Figure 6c according to Equation (2). The value of t/Q_t_ and the adsorption time was linear fitted. The R^2^ of the pseudo-second-order fittings for FTFM ranged from 0.991 to 0.998, as presented in Table 2, which was significantly higher than that of the pseudo-first-order fitting. Moreover, the equilibrium adsorption capacity calculated from the slopes by the pseudo-second-order fitting (Q_e2_) ranged from 1.09 ± 0.012 to 12.61 ± 0.016 mmol/g, which was consistent with the experimental values. However, it was not scientific when only the coefficient of determination and the calculated adsorption capacity were considered in the metal ion adsorption model. In addition, the assessment of data correlation quality (AARD) should be calculated and compared (reference). The AARD was calculated according to Equation (3), where $\hat{y}$_i_ = ln(Q_e_-Q_t_) and y_i_ = lnQ_e_-k_1_t for the pseudo-first-order fitting, and $\hat{y}$_i_ = t/Q_t_ and y_i_ = t/Q_e_ + 1/k_2_Q_e_^2^ for the pseudo-second-order fitting. The AARD_1_ and AARD_2_ were compared by calculating the R_(1-2)_ value according to Equation (4), which is presented in Table 2. The AARD_1_ ranged from 118.86% to 421.76%, two orders higher than AARD_2_, which ranged from 1.15% to 3.91%. The R_(1-2)_ values were equal to or greater than 0.99, indicating that the silver ion adsorption of FTFM followed the pseudo-second-order model. Thus, the adsorption process on the FTFM surface was dominated by the chemical chelation, and a single layer of silver was supposed to be generated on the petals after the adsorption.
(1)$$*\;\ln\left(Q_{e}-Q_{t}\right)=lnQ_{e}-k_{1}t$$
(2)$$t/Q_{t}=t/Q_{e}+1/k_{2}Q_{e}^{2}$$
(3)$$\mathrm{AARD}=\frac{1}{N}\sum_{i=1}^{N}\left|\frac{\hat{y}i-y_{i}}{y_{i}}\right|$$

** R_(1-2)_^2^ was calculated to compare the correlation quality between pseudo-first order and pseudo-second-order fitting according to the following equation:(4)$$R_{\left(1-2\right)}^{\;2}=1-\frac{\sum_{i=1}^{N}\left|y_{i\left(2\right)}-\hat{y}i_{\left(2\right)}\right|}{\sum_{i=1}^{N}\left|y_{i\left(1\right)}-\hat{y}i_{\left(1\right)}\right|}$$

The isothermal study of the silver ion adsorption was applied to the FTFM calculating the maximum adsorption capacity (Q_max_) for the parallel comparison. As presented by the kinetic study, the adsorption reached equilibrium when the time exceeded 240 min. However, the adsorption capacity at the equilibrium was affected by the initial concentration of AgNO_3_ (C_0_), and the relationship between Q_e_ and C_0_ was demonstrated by Figure 7a. The Q_e_ kept increasing with the initial concentration, and the corresponding value for the solid microsphere was significantly lower than that of the flower-like microspheres due to the huge difference in the specific surface area. The Q_e_ of the FTFM samples with different thiourea/formaldehyde molar ratios kept at the same level when the C_0_ was 20 and 40 mmol/L, however, changed significantly when C_0_ exceeded 60 mmol/L. When the C_0_ was 100 mmol/L, the Q_e_ increased significantly with the molar ratio between formaldehyde and thiourea, which was consistent with the structure characterization results. The residual silver ion concentration at the adsorption equilibrium (C_e_) was calculated according to the UV vis spectrum. All of the adsorption processes were fitted by the Langmuir adsorption model (Equation (5)). The C_e_ and C_e_/Q_e_ were linear fitted and presented by Figure 7b. The R^2^ of all the FTFM samples was larger than 0.99, and the value for the solid microspheres was 0.959. The fitting results indicated that the silver ion adsorption process presented in this work followed the Langmuir model. The Q_max_ for all of the samples was calculated by the reciprocal of the linear fitting slops demonstrated in Table 3. The Q_max_ of S_2_T_1_F_4_M was only 1.32 ± 0.16 mmol/g, and the corresponding value for FT_1_F_0.5_M was 10.92 ± 0.46 mmol/g, which was around an order higher than that of the solid microspheres. Q_max_ kept increasing with the molar ratio between formaldehyde and thiourea, which was consistent with the BET specific area results. The maximum Q_max_ in this work reached 18.17 ± 1.28 mmol/g for FT_1_F_4_M, which was 13.8 times that of the thiourea–formaldehyde solid microspheres.
(5)$$\frac{c_{e}}{Q_{e}}=\frac{c_{e}}{Q_{\max}}+\frac{1}{{\mathrm{bQ}}_{\max}}$$

As shown in Appendix A, the Ag^+^ adsorption capacity of FT_1_F_4_M prepared in this work was significantly higher than the published works about thiourea–aldehyde Ag^+^ adsorption resin, which are presented in the Appendix A.

## 4. Conclusions

Thiourea–formaldehyde resin silver adsorbent with flower-like microspheres micro-morphology was prepared and characterized. The silver adsorption capacity was greatly enhanced by the flower-like microsphere structure by increasing the specific areas rather than the chemical formulation optimization. The flower-like microsphere structure was controllable by turning the reaction conditions, and the relationship between the morphology and the adsorption ability was researched. The adsorption kinetic was explored, indicating that the adsorption process on the FTFM surface was dominated by the chemical chelation, and a single layer of silver was supposed to be generated on the petals after the adsorption. The results indicated that the specific surface area of FTFM was up to 18.97 m^2^/g, which was 55.8 times than that of the solid microspheres control. The silver ion adsorption ability was greatly enhanced, reaching 7.95 ± 0.396 mmol/g in AgNO_3_ aqueous solution (0.1 mol/L, 20 min), which was 10.9 times that of the control. The adsorption kinetic study indicated that the correlation quality of the pseudo-second-order model was significantly higher than that of the pseudo-first order model, where R_(1-2)_ was greater than 0.99. The Q_e_ of FT_1_F_4_M calculated by the pseudo-second-order model was 12.61 ± 0.016 mmol/g, 11.6 times that of the thiourea–formaldehyde solid microspheres control. The isothermal study was applied to the silver ion adsorption, and the C_e_ and C_e_/Q_e_ was linear fitted presenting good correlation quality. The calculated Q_max_ of FT_1_F_4_M was 18.17 ± 1.28 mmol/g, which was 13.8 times that of the control. The strategy presented by this work provided a rapid, simple, and effective way of enhancing the silver adsorption capacity of thiourea–formaldehyde resin with the mechanism clarified. Its high absorption efficiency, convenient preparation strategy, and low cost recommend FTFM bright for further use in industrial applications.

## Data Availability

Data is contained within the article or Appendix A. The data presented in this study are available in Appendix A.

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
