# Peer review of "Flower-like Thiourea–Formaldehyde Resin Microspheres for the Adsorption of Silver Ions"

_polymers, 2023, doi:10.3390/polym15112423_

Round 1

Reviewer 1 Report

The work is good but needs minor revision.

1.      The authors need to reduce the length of the abstract.

2.      Rewrite the keywords.

3.      The authors should give justification to prove the novelty of this research.

4.      The authors must include the EDAX analysis of the prepared materials.

5.      In Figure 4 (c-e): SEM and elemental mapping images are not having a scale bar.  Check it carefully.

6.      In Figure 5: Y-axis has no unit; check it carefully.

7.      The authors need to discuss the reusability of the prepared catalysts with evidence of FTIR or SEM analysis (after recyclability).  

8.      The authors must include the comparison table of silver ion adsorption efficiency results with previously published results.

9.      Novelty should be highlighted in the conclusion part and also reduce the length of the conclusion part.

10.  Improve the figure captions and add more details.

11.  Figure quality should be improved.

12.  More typos in the manuscript; double-check it thoroughly.

13.  English expressions can be further improved.

English expressions can be further improved.

Author Response

Dear reviewer:

Thanks for your suggestion. We read carefully and replied one by one, see the following documents.

Kind regards,

Yuhan Li

Reviewer 2 Report

The authors executed their idea and concept well but I would like to know the following details for further.

1. "Adsorption Properties of Silver Ions on Thiourea-Formaldehyde Resin" Advanced Materials Research Vol. 868 (2014) pp 459-462. This is one of the author's published works which is also significantly discussed regarding similar strategies to this submitted work. 

Please explain what the morphology of the materials is how significant and what is the significant difference between these two studies.

2. Please elaborate on their advantages in this field (in the introduction section ).

3. Figures quality is poor please revise them.

Author Response

Dear reviewer:

Thanks for your suggestion. We read carefully and replied one by one, see the following documents.

Kind regards,

Yuhan LI

Round 2

Reviewer 2 Report

The authors revised the manuscript and responded to the comments accordingly. All the mentioned and required changes are done in the revised manuscript and improved.